# Identifying Financial Crises Using Machine Learning on Textual Data

**Mary Chen** [1] , **Matthew DeHaven** [2], **Isabel Kitschelt** [3], **Seung Jung Lee** [3,*] **and Martin J. Sicilian** [4]

1   Federal Reserve Bank of Boston, Boston, MA 02210, USA
2   Department of Economics, Brown University, Providence, RI 02912, USA
3   Board of Governors of the Federal Reserve System, Washington, DC 20551, USA
4   Stanford Law School, Stanford, CA 94305, USA
*   Correspondence: seung.j.lee@frb.gov

**Abstract:** We use machine learning techniques on textual data to identify financial crises. The onset of a crisis and its duration have implications for real economic activity, and as such can be valuable inputs into macroprudential, monetary, and fiscal policy. The academic literature and the policy realm rely mostly on expert judgment to determine crises, often with a lag. Consequently, crisis durations and the buildup phases of vulnerabilities are usually determined only with the benefit of hindsight. Although we can identify and forecast a portion of crises worldwide to various degrees with traditional econometric techniques and using readily available market data, we find that textual data helps in reducing false positives and false negatives in out-of-sample testing of such models, especially when the crises are considered more severe. Building a framework that is consistent across countries and in real time can benefit policymakers around the world, especially when international coordination is required across different government policies.

**Keywords:** financial crises; machine learning; natural language processing

*I believe there is no deep difference between what can be achieved by a biological brain and what can be achieved by a computer. It, therefore, follows that computers can, in theory, emulate human intelligence—and exceed it.*

—Stephen Hawking.

## 1. Introduction

In this paper, we use machine learning techniques on textual data to identify and predict financial crises. The academic literature and the policy realm rely mostly on expert judgment to determine financial crises.[1] Consequently, the identification of crisis periods and the buildup phases of vulnerabilities are usually determined by experts with the benefit of hindsight. This implies that various data on financial and banking crises are slow to update; in many cases, updates only occur after many years (Reinhart and Rogoff (2009); Laeven and Valencia (2013)) or not at all (Romer and Romer (2017), Baron et al. (2020)).

In our analysis, we build an indicator that signals in real time the entire period during which a particular country is in a crisis. Due to the limitations of quantitative variables available for a wide range of countries, we use machine learning techniques on the following textual data: reports from official international organizations (the Organization for Economic Co-operation and Development (OECD) and International Monetary Fund (IMF)) and articles from the media (Refinitiv, Machine Readable News (MRN), Reuters Daily News Feed (RDNF)). These data sources help us to develop an indicator that increases quickly and stays elevated for the entirety of the crisis period. Although a decent portion of certain types of financial crisis periods can be identified with traditional econometric

techniques using readily available market data, we find that textual data significantly helps in reducing false positives and false negatives in out-of-sample testing of such models, especially when the crises are considered more severe. Moreover, our model can even detect nontraditional forms of financial crises; for example, it is able to determine that the recent COVID-19 pandemic was a financial crisis in the United States.

Real-time identification of financial crises is important for policymakers in conducting macroprudential policy and crisis management as well as for monetary policy and fiscal policy. In the context of macroprudential policy, an elevated reading can provide valuable inputs into decisions regarding when and for how long to release the counter-cyclical capital buffer (CCyB) and when to begin increasing it again, for example. The CCyB is an additional capital requirement levied on banks to counter procyclicality in the financial system. In many countries, CCyBs are activated when vulnerabilities in the financial system are high; thus, banks have to build up capital in good times. The buffer can be released when risks materialize or a crisis is realized, as was the case for some countries during the height of the COVID-19 pandemic. In this way, the banks would have incentives to be more prudent with their lending in good times, while macroeconomic benefits can be realized from banks' ability to maintain credit flows in bad times. In the context of monetary and fiscal policies, the last thing policy makers would want to do is to be more hawkish with monetary policy or stringent with fiscal policy when a financial crisis has not yet concluded. Premature monetary tightening or austerity policies can lead to significantly weaker real activity than otherwise, and can possibly prolong a crisis. Building a framework that is applicable to many countries in the world can potentially help with international coordination of various policies if necessary. In other words, having a consistent way to identify crises across countries in such circumstances would be crucial in dealing with crises in an internationally concerted manner.

We present a number of techniques to uncover the black-box nature of the machine learning models we develop, which helps us to understand the nature of the identified financial crises. In this case, the main inputs into the models, which are words, provide insights into what is driving the results. For example, we can clearly see that the Global Financial Crisis was more of a banking crisis, as opposed to the COVID-19 pandemic, which had to do with financial hardships more generally.

Finally, we forecast financial crisis periods using similar models and discuss how to interpret the different results obtained when using different types of text as input. For example, models using media articles are better at forecasting financial crises than models using reports from official institutions; however, official reports have the upper hand when it comes to identifying crises in real time. This may imply that media, with its greater readership, may have more instigative properties, as opposed to the purely descriptive characteristics of text associated with official institutions. This is consistent with the narrative view described in Shiller (2017), which emphasizes that popular stories can affect individual and collective economic behavior.

After starting with a literature review in Section 2, we describe the data and model in Sections 3 and 4, respectively. Sections 5 and 6 respectively describe our machine learning model and show our results for nowcasting and forecasting financial crises, with an emphasis on out-of-sample performance metrics and explainable artificial intelligence. We also discuss why different types of text can be useful in identifying and forecasting financial crises, respectively. In Section 7, we provide our conclusions.

## 2. Literature

Our research contributes to the literature on predicting and detecting financial crises and, more broadly, on using machine learning and textual data to nowcast and forecast aggregate macroeconomic and financial conditions.

As for predicting and identifying crises, Drehmann and Juselius (2014), Aikman et al. (2017), Lee et al. (2020), and Cesa-Bianchi et al. (2019) provide a framework for understanding the financial vulnerabilities that lead to financial crises. Brave and Butters

(2012) provide a way to understand how financial conditions can forecast financial crises. The literature points to many different types of vulnerabilities increasing prior to financial crises, especially when they end up being systemic. As for directly maximizing predictive power using machine learning on quantitative data, Alessi and Detken (2018), Bluwstein et al. (2020), and Fouliard et al. (2022) provide analyses to show how machine learning can be very powerful in predicting financial crises. The context of these papers is mainly related to understanding the factors that lead to financial crises, as opposed to identifying different types of crises.

All of these papers focus on the onset date of financial crises as the main dependent variable, and ignore ongoing "crisis" states for significant periods of time. In fact, relevant quantitative indicators in these models behave in a way such that either indexes spike at the onset of crises (or at other near-crisis periods of market disruptions), or gradually increase and then peak around the onset of crises and fall sharply afterwards. Figure 1 illustrates how these types of metrics behave for the United States juxtaposed with financial crisis periods according to Romer and Romer (2017) with a certain level of severity. The LPS financial vulnerability measure from Lee et al. (2020) becomes elevated prior to crises and falls sharply afterwards, as it is an indicator of vulnerabilities building up in the financial system. Likewise, when looking at a crisis prediction model based on realized volatility, which is similar to metrics that are used in many financial stress indexes, sharp increases occur more frequently than financial crises, meaning that the metric has a tendency to provide false positives in detecting financial crises.

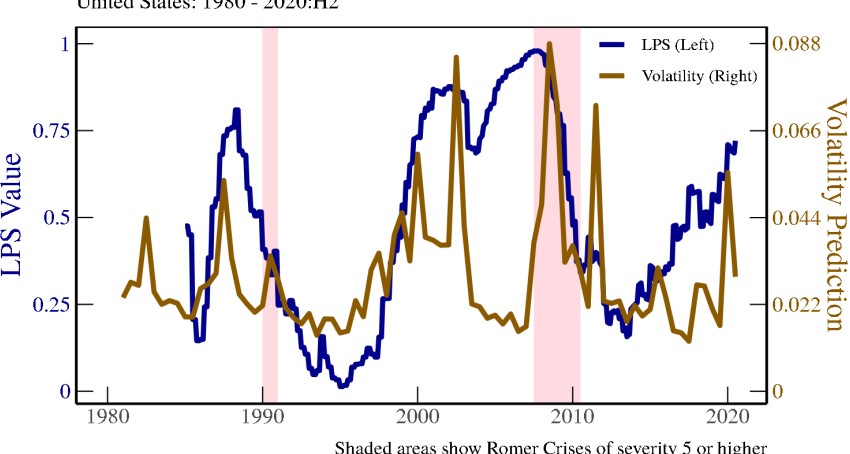

**Figure 1. Different Financial Stability Metrics.** *Note:* The figure shows the Lee–Posenau–Stebunovs (LPS) vulnerability index for the United States (lfrom Lee et al. (2020)) and predictions from a realized volatility model calculated from US stock markets (similar to the measure used in Duprey et al. (2017) and Danielsson et al. (2018)). The shaded regions indicate periods of crisis severity 5 or higher according to Romer and Romer (2017). These two metrics illustrate how current quantitative indicators of financial stability cannot provide consistent information regarding the onset of crisis states or the length of crises.

As for other studies identifying crises, Duprey et al. (2017) provide a method using Markov switching and a threshold-based vector autoregressive model using financial stress indexes and industrial production data to identify financial crises for European Union countries. Laeven and Valencia (2013), meanwhile, use a combination of narrative and quantitative threshold approaches to identify financial crises; however, these take a significant amount of time to update. Baron et al. (2020) provide a way to identify banking crises in real time using bank equity price data, although this may not detect financial crises that are not banking-related. Romer and Romer (2017), on the other hand, define the severity of financial distress for 24 OECD countries from 1967 to 2012 by reading through

each country's OECD Economic Outlooks. This narrative determination of crises is linked to declines in output afterwards, with the variation in the declines driven by the severity and persistence of the financial distress itself. Our study incorporates the approach used in Romer and Romer (2017), then applies machine learning to the OECD Economic Outlooks and other textual data based on the training sample (of crises) constructed by Romer and Romer (2017) in order to develop a real-time indicator of financial crises.

Indeed, using textual analysis to predict and identify various macroeconomic and financial aggregates has become more and more popular in recent years. For example, Angelico et al. (2022) use Twitter feeds to understand inflation expectations, while Kalamara et al. (2022) use U.K. newspaper articles to forecast a very wide set of aggregate macro-financial variables such as GDP growth, inflation, and financial vulnerability measures. This new literature points to an expanded set of information that can be used to help understand various macro-financial aggregates. In addition, studies such as Cerchiello et al. (2017) nowcast financial distress at the individual bank level using textual data. As with our work, all of these papers provide further insight into the determination of macro-financial variables through text.

### 3. Data

This section provides an overview of the crisis data we use for our dependent variables. We describe the independent variables used for crisis identification and prediction, including textual data, market data, and credit data.

### 3.1. Crisis Data

While we use a variety of crisis data in our analysis, we rely on two main sources for crisis determination. First, Romer and Romer (2017) provides an ideal set of crisis definitions for our textual analysis. They define crises with a narrative approach focused on the OECD Economic Outlook—a roughly 2000-word quantitative and contemporaneous document—which has been published twice a year since 1967 for different OECD countries. Although mostly focused on the economic environment and real-side forecasts, these documents are read for signs of a rise in the cost of credit intermediation and adverse effects on real activity for individual countries. As in Figure 2, they develop definitions for five types of financial distress increasing in severity: credit disruption, minor crisis, moderate crisis, major crisis, and extreme crisis. Each category is assigned a minus, normal, or plus. Thus, their final numerical scale ranges from 0 to 15. This constitutes a more continuous measure of crises. Their crisis definitions are provided below:

- **Credit Disruptions**: While the OECD perceived strains in financial markets, funding problems, or other indicators, it did not believe these to have any macroeconomic impacts.
- **Minor Crisis**: There is a perception of significant problems in the financial sector, along with a belief that these problems are affecting the credit supply and/or overall economy in a nontrivial way and are not limited to a minor part of the economy, yet the impact is not large enough to be damaging to the economy's overall prospects.
- **Moderate Crisis**: Widespread and severe problems in the financial sector are central to the economy as a whole, yet not serious enough to be described as the financial system seizing up entirely.
- **Major/Severe Crisis**: Romer and Romer (2017) look for the terms "crisis", "dire", "grave", "unsound", and "paralysis" in reference to the financial system in order to classify this level; references to major government interventions contribute to a severe rating as well.

In our analysis, we focus on whether crises are at least minor crises, using severity of a five or greater as a benchmark. When restricting our analysis to major/severe crises, there are too few observations to conduct a meaningful analysis.

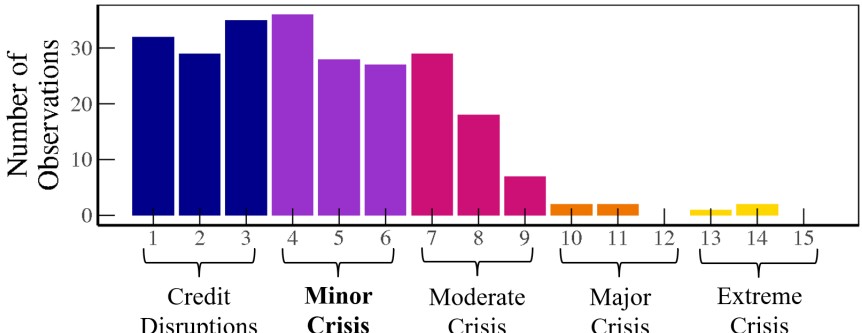

**Figure 2.** Romer and Romer (2017). *Note:* The figure shows the number of observations for each crisis severity level (1–15) and type defined by Romer and Romer (2017): credit disruption, minor crisis, moderate crisis, and major/severe crisis. The most frequent crisis level is 4, and the majority of crises are categorized as credit disruptions or minor crises.

Our second source of crisis data is from Laeven and Valencia (2013). We focus on banking crises, as this provides a higher (monthly) frequency of crisis dates available for a broader set of countries. A banking crisis is defined as an event that meets two conditions: (1) significant signs of financial distress in the banking system, and (2) significant banking policy intervention measures in response to significant losses in the banking system. The first year that both criteria are met is the year when the crisis becomes systemic. When losses in the banking sector and/or liquidations are severe, the first criterion is a sufficient condition. Losses are severe when either (i) a country's banking system exhibits significant losses resulting in a share of nonperforming loans above 20 percent of total loans or bank closures of at least 20 percent of banking system assets, or (ii) fiscal restructuring costs in the banking sector are sufficiently high, exceeding 5 percent of GDP. Policy interventions in the banking sector are considered significant if at least three of the following six measures have been used: (1) deposit freezes and/or bank holidays, (2) significant bank nationalizations, (3) bank restructuring fiscal costs (at least 3 percent of GDP), (4) extensive liquidity support (at least 5 percent of deposits and liabilities to nonresidents), (5) significant guarantees put in place, and (6) significant asset purchases (at least 5 percent of GDP).

### 3.2. Volatility and Credit Data

Our benchmark quantitative model relies on realized volatility from stock markets worldwide from the GFDatabase from Global Financial Data, Inc. (San Juan Capistrano, CA, USA), similar to the measure used in Duprey et al. (2017) and Danielsson et al. (2018). These data are available for all 24 OECD countries in our main sample going back to at least 1981 for 14 countries.[2] With the exception of two countries, these data reach back to at least 1988. This points to the lack of widely available financial market data for long time series across multiple countries. Nonetheless, this simple model allows us to look at a wide set of countries for a long time period and permits out-of-sample testing. More importantly, market disruptions usually show up in such volatility measures, and feature predominantly in various indicators of financial stress or financial conditions. Throughout the rest of this paper, we refer to this model as the "Volatility Model". In addition to this broadly-available market-based measure, we look at the credit-to-GDP gap from the Bank for International Settlements (BIS), as used in Drehmann and Juselius (2014). These data are available for all 24 OECD countries except Iceland and Luxembourg. With the exception of the data for Turkey, which start in 1996, the data for other countries reach back to at least 1981. The credit-to-GDP gap is considered a financial vulnerability metric that has a more forward-looking element when it comes to financial stress. However, when identifying crisis periods these models perform poorly, and we do not emphasize such results in this paper; however, we do show them later on when we consider forecasting exercises.

### 3.3. Textual Data

As for textual data, we rely on a few sources that are both contemporary and publicly available.[3] We mostly rely on the OECD Economic Outlook, as in Romer and Romer (2017), as described above. The textual data herein provide a large panel of textual data beginning in 1967 for several OECD countries published in a regular semiannual frequency. The OECD Economic Outlooks are available for all 24 OECD countries in our sample starting in 1981. The OECD Economic Outlooks are prepared by the OECD Economics Department, and cover economic trends and prospects for the next two years, including output, employment, government spending, etc. They are usually no longer than ten pages, and typically provide a good synopsis of where a country's economy is heading. In particular, when the financial sector is mentioned this is a sign that it is weighing on real economic activity, which is at the core of how experts identify financial crises.

Next, we consider the media-based Refinitiv RDNF. The RDNF begins in 1996, and the frequency is to the millisecond; we aggregated this to the monthly frequency for each particular country. Beginning with about 40 million articles, we filter out sports-related articles and only keep economics and finance related articles written in English, which brings down the count to approximately 20 million. We next use the R package *newsmap* (Watanabe 2018), a semi-supervised Bayesian model, to tag articles to specific countries. Figure 3 provides an overview of the final article counts for each country in log scale. The lighter colors represent the countries with a relative abundance of articles in the dataset. Only countries in Africa and the Middle East have relatively few non-sports articles of the types used in our textual analysis. Although these data begin in 1996, we are able to expand the number of countries in our analysis to 62 based on further cleaning of the data.

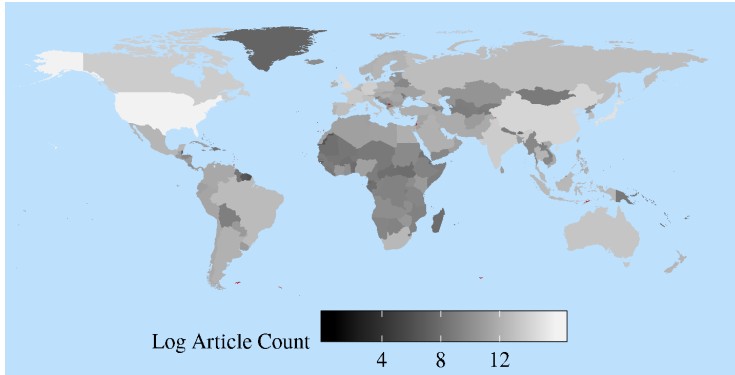

**Figure 3. RDNF Articles by Country**. *Note:* Of the approximately 40 million initial RDNF articles, after filtering we obtained approximately 20 million articles related to economics and finance written in English. We kept articles containing economics and finance terms and dropped articles containing sports terms, then implemented the R package *newsmap* (Watanabe 2018), a semi-supervised Bayesian model, to tag articles to specific countries. The figure shows the country coverage of RDNF log-transformed article counts from January 1996 to July 2021. The lighter colors represent countries with a relative abundance of articles in the dataset; only countries in Africa and the Middle East have relatively low coverage of economics and finance articles.

Finally, we include IMF Article IVs from the IMF in our analysis, which is available for a broad set of countries, though the frequency is more scattered. Similar to the OECD data, these documents follow a consistent pattern and cover different aspects of financial crises if they occur in a particular country. IMF Article IVs have a long time series, and can be potentially used for many more countries; we download Article IVs for 39 countries reaching back to the early 1980s. Similarly, Romer and Romer (2017) crosscheck their narratives based on the OECD Economic Outlooks with the IMF Article IVs.

Table 1 provides a summary of all the data sources, including the number of countries represented (after cleaning) and the time period in which analysis can be conducted. It

is immediately apparent that the OECD sample has the smallest number of countries but spans the longest time period.

**Table 1.** Summary statistics of data sources.

| Dataset | Number of Countries Available | Number of Countries Used | Time Period Available | Base Frequency |
|---|---|---|---|---|
| R&R Crises | 24 | 24 | 1967–2012 | biannual |
| L&V Crises | 118 | 62 | 1976–2017 | monthly/annual |
| OECD Text | 44 | 24 | 1967–2020 | biannual |
| RDNF Text | ∼238 | 62 | 1996–2020 | minutely |
| IMF Text | ∼111 | 39 | 1983–2020 | annual |

*Note:* The table shows country coverage, historical coverage, frequency of Romer and Romer (2017) crisis and Laeven and Valencia (2013) banking crisis data, and OECD, RDNF, and IMF textual data. We use Romer and Romer (2017) crisis data for 24 countries from 1967–2012 at a biannual frequency and Laeven and Valencia (2013) banking crisis data from 62 countries from 1976–2017. The OECD textual data are from 1967 for 24 countries. The RDNF textual data begin in 1996 with to-the-millisecond frequency, which we aggregate to monthly. The IMF Article IVs have the longest time series from the early 1980s, and are available for about 40 countries at annual frequency.

Basic sentiment analysis can provide insight into how the different documents could be potentially useful. Figure 4 plots the range of sentiment scores from each source text generated with a word dictionary developed by Correa et al. (2017) using basic sentiment scores based on the number of positive minus negative words over the total number of positive and negative words. These plots suggest that the documents contain promising information that can be quantified using text feature extraction, which may help in detecting financial crises in real time and predicting future crises. Analyzing the documents simply using sentiment scores, which is one of the most basic uses of text features, demonstrates that the documents convey important patterns regarding crises, most obviously seen by the dip in sentiment for all three texts during the 2008 period coinciding with the Global Financial Crisis. Furthermore, the COVID-19 pandemic appears to have brought down sentiment in the beginning of 2020 in all of our text sources as well.

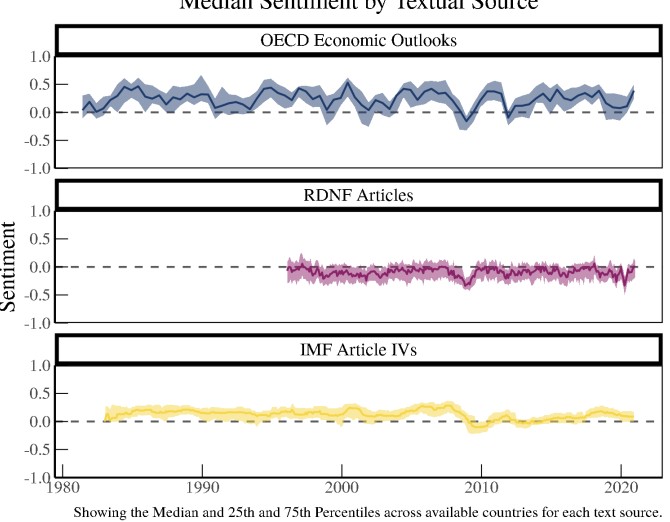

Figure 4 showing "Median Sentiment by Textual Source" with panels for OECD Economic Outlooks, RDNF Articles, and IMF Article IVs.

Showing the Median and 25th and 75th Percentiles across available countries for each text source.

**Figure 4. Median Sentiment Scores Using Different Texts**. *Note:* This figure shows Correa et al. (2017) Financial stability (FS) sentiment scores for OECD Economic Outlooks, RDNF Articles, and IMF Article IVs using basic sentiment scores based on the number of positive words minus negative words over the total number of positive and negative words. Financial stability sentiment scores demonstrate that the documents convey important patterns regarding crises, most obviously seen from the dip in sentiment for all three texts during the 2008 period coinciding with the Global Financial Crisis.

## 4. Machine Learning on Textual Data

In this section, we briefly describe the data cleaning and setup, before delving into exactly how we conduct our various machine learning exercises. Then, we describe the metrics we use to determine out-of-sample performance in determining and forecasting financial crises.

Before running our machine learning models, we clean the textual data and format them for use as inputs to our models in the form of document feature matrices. Prior to cleaning any text, we create an "OECD dictionary" from the OECD Economic Outlook text source by removing the most and least frequently used terms in the OECD Economic Outlooks from 1967 to 1980. Appendix B explains the process in greater detail. This allows us to follow 881 terms (both unigrams and bigrams) existing throughout the sample, thereby helping to minimize look-ahead bias. For example, we want to avoid crisis-specific words such as "Asian Crisis" or "mortgage-backed securities" to drive our results in the out-of-sample analysis.

After constructing the OECD dictionary, Figure 5 shows the procedure used to process OECD text using an example sentence in the red box: "The expansion slowed down considerably in the second half of the year, influenced by the weaker trend in the United States and tighter policies at home". We start by tokenizing the text such that each word is separated. We then remove stop words such as "the", "in", and "by" and lemmatize the text; words such as "expansion" and "expanded" are all standardized to "expans". Using the cleaned text, we create an $m \times n$ document feature matrix, with $m$ number of documents and $n$ number of words (or terms), to record the frequency of words within the text. We then compare the words in the document feature matrix against the OECD dictionary, keeping only those features in the OECD dictionary. In our example, the blue box displays the output, which is an $m \times 881$ document feature matrix with the 881 terms from the OECD dictionary. As a final step, we then normalize this document feature matrix by the total number of terms in a particular OECD Economic Outlook. For simplicity, the procedure shown in in Figure 5 does not show bigrams; however, our text processing includes both unigrams and bigrams. We construct this $m \times 881$ document feature matrix for each text source (OECD, RDNF, IMF Article IVs) for use as the input variables of the machine learning models.

Next, we describe how we utilize the textual information we collect in a machine learning model. We split the time dimension into a training set and testing set to avoid data leakage. We decide to split our data at the end of 2004, meaning that our out-of-sample prediction results are based on how well our model is able to identify/predict crises primarily during the GFC period. In other words, we use data from 1981 to 2004 to train the data for creating and tuning the models. Tuning of the models is achieved by splitting the training data into multiple folds, again along the time dimension. We create ever-increasing "validation" sets of the data, always predicting one year forward, training on all of the data to that point. In this way, we ensure that the time-series nature of the data is respected. By comparing performance on these validation sets, we select optimal hyperparameters and train a final model on the whole training data using those parameters.

The main machine learning models used in this research are the Support Vector Machine (SVM) and Random Forest; in addition, we tried GLMNET (based on Elastic Net, Ridge, and Lasso), Neural Net, Adaptive Boosted Forest, Extreme Random Forest, and others. Consistent with the literature on machine learning for categorization purposes, SVM and Random Forest are the most efficient as classification models (see Kumar and Thenmozhi (2006)); thus, many of the results we show here are based on the averaged results of these two models. Regardless of which of these models is used, the results are similar.

SVM is a supervised machine learning algorithm that classifies data by constructing hyperplanes that separate categories with the maximal margin. We use the radial basis function kernel in our implementation of SVM. The Random Forest algorithm is another supervised machine learning classification algorithm which uses decision trees to return

the most probable prediction. Kumar and Thenmozhi (2006) describes the exact equations for the two methods and demonstrates their usage in a forecasting exercise with financial data. Here, we use the *caret* package (Kuhn 2022) in R to implement the SVM and Random Forest classifiers.

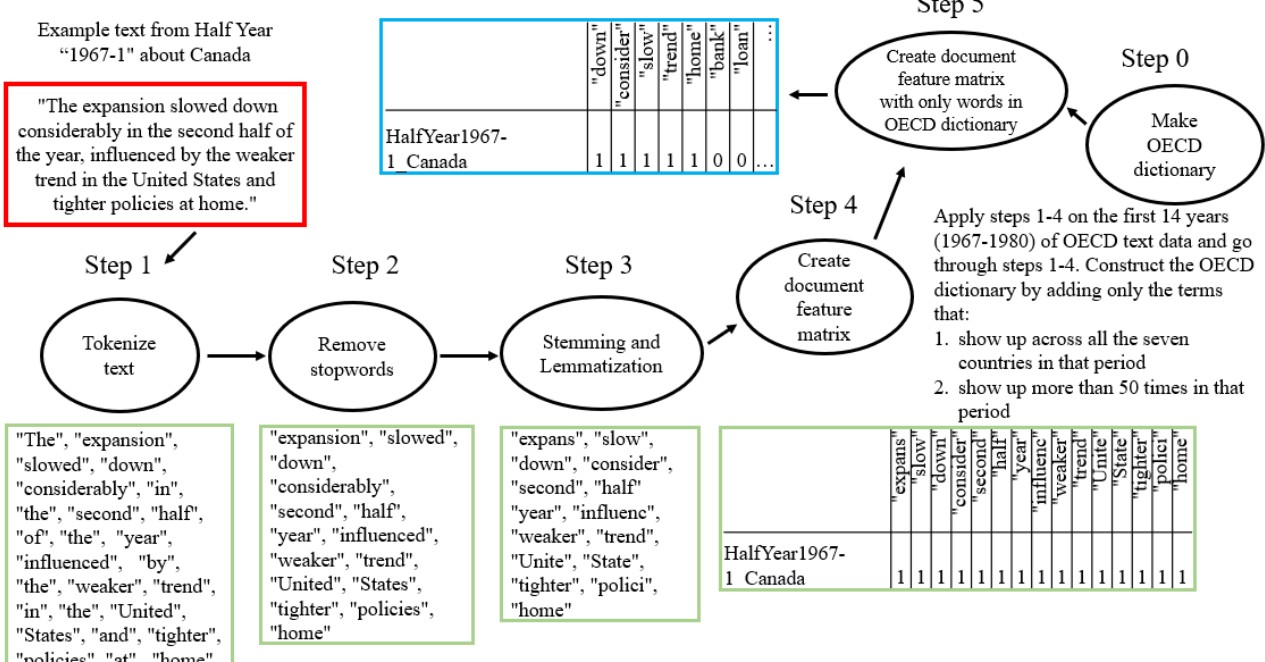

**Figure 5. Text Data Setup**. *Note:* This figure describes the process of cleaning the textual data, demonstrating the process on a sentence from Canada's entry in the OECD Economic Outlook, Volume 1967, Issue 1. As a preliminary step, we create an OECD dictionary based on OECD text data by removing the most and least frequently used terms in the OECD Economic Outlook text source from 1967 to 1980; this leaves us with 881 terms, comprising unigrams and bigrams (for simplicity, the figure only shows unigrams) that exist throughout the sample, thereby minimizing look-ahead bias. For the documents of each of the text sources (OECD, RDNF, IMF Article IVs), we tokenize the text, remove common English stop words, stem and lemmatize the text, and create a document feature matrix. We then create a subset of the words in the OECD dictionary to construct our final document feature matrix for each text source.

Using our testing data from 2005 to 2012 and looking at the area under the Receiver Operating Curve (ROC), or AUROC statistics, we can assess the out-of-sample performance. In addition to this "chunk" method, in which we train the model up to the end of 2004 and then look at out-of-sample properties after 2004, we provide results for the "expanding horizon" model, where re-estimation is performed while moving along the time dimension. In the context of identifying financial crises, and assuming that there is a stable relationship, either of these methods should work. Assuming that there is a time-varying relationships between text and financial crises, the rolling-window method may be superior. As it turns out, the rolling-window results are very similar to the expanding sample results; therefore, we only report the "chunk" and "expanding" results.

In order to check whether text data add value to out-of-sample predictions, we use the baseline logit model based on realized volatility calculated from daily stock return data. We then compare the machine learning models with these logit models mentioned above. To predict financial crises, we also report the results of the logistic regressions using the

credit-to-GDP gap as the regressor. In essence, the logistical regressions we compare our machine learning models to are defined as follows:

$$Pr(Y_{it} = 1 | X_{it}) = \frac{exp(\beta_0 + \beta_1 X_{it})}{1 + exp(\beta_0 + \beta_1 X_{it})} \tag{1}$$

where $Y_{it}$ is a crisis dummy and $X_{it}$ is either the sentiment measure, realized volatility measure, or the credit-to-GDP gap measure for country $i$ at time $t$.

## 5. Nowcasting Results

### 5.1. Identifying Romer and Romer Crises Using OECD Text

Figure 6 describes the variable importance for our first exercise, which trains machine learning models to identify Romer and Romer (2017) minor or more severe crises (score of 5 or more) using only OECD Economic Outlooks from 1981–2004. The variable importance is based on the percentage area under the receiver operating characteristic curve (AUROC) gains from including a particular word in the model. It can be seen that the words "bank", "loan", and "financial", along with their various interactions, provide valuable insights into detecting whether or not there is a crisis, regardless of which model is being examined. Indeed, when the OECD Economic Outlooks mention these words and terms, it is usually due to the fact that problems in the financial sector are weighing on real economic activity and the outlook. Other terms, such as "weakness", can be useful in identifying financial crises from the OECD Economic Outlooks as well. The variable importance charts show similarities in our main machine learning models, namely, SVM Radial and Random Forest.

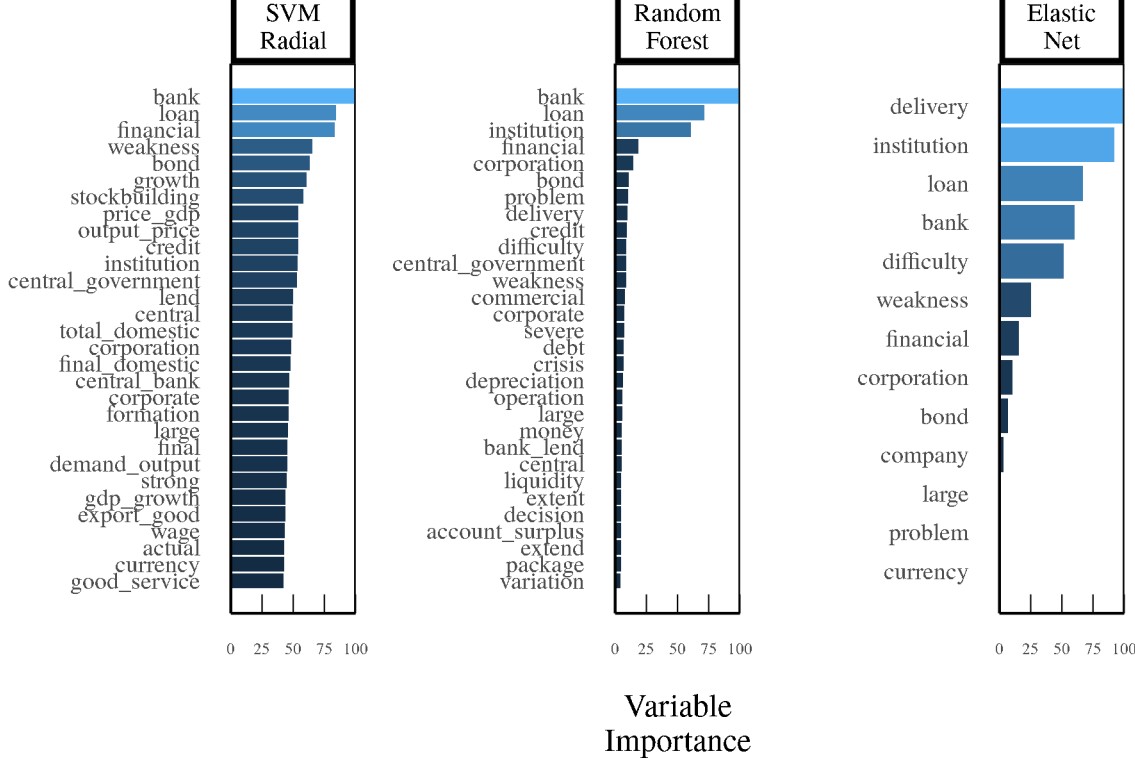

Variable Importance

**Figure 6. Variable Importance for R&R Minor Crises on OECD Text**. *Note:* This figure shows variable importance for three machine learning models based on Romer and Romer (2017) 5+ severity crises trained on OECD text from 1981–2004. Variable importance is based on the percentage gain of the area under the curve (AUC) when including a particular word in the model. Words such as "bank" and "loan" provide valuable insight for the purpose of detecting crises. The variable importance charts between our main machine learning models (SVM Radial and Random Forest) are similar.

As shown in Figure 7, out-of-sample identification for the receiver operating characteristic (ROC) curves (from 2005 to 2012) using models trained with the "chunk" method (i.e., trained on text from 1981–2004) suggests that the average of the text-based SVM Radial and Random Forest approaches, henceforth referred to as the Average Text Model (with an area under the curve (AUC) of 0.91), vastly outperforms the volatility-only model (Volatility Model with an AUC of 0.80), which in turn outperforms the sentiment score-based model (Sentiment Model with an AUC of 0.76). When we average the probability estimate outputs from all of the models (i.e., Average Text Model, Volatility Model, and Sentiment Model), henceforth referred to as the Averaged Model, we see further improvements in the AUC, although the improvement over the Average Text Model is very small. Nonetheless, an AUC over 0.92 is considered very high in the literature on identifying and predicting financial crises.

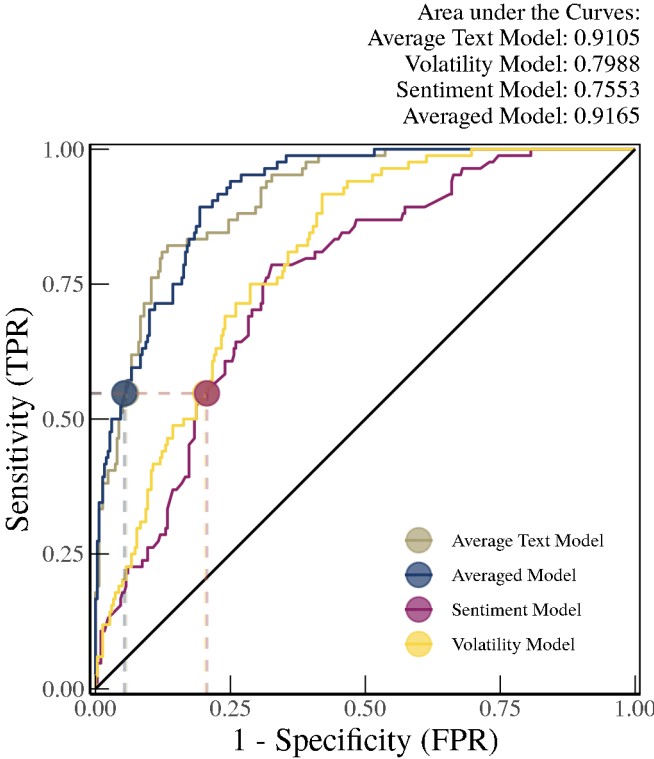

**Figure 7. Results for R&R Minor Crises on OECD Text**. *Note:* This figure shows the receiver operating characteristic (ROC) curves for our out-of-sample period (2005–2012) for models trained on the period 1981–2004 using OECD text based on Romer and Romer (2017) 5+ severity. The results show that the Average Text Model (based on the average of our SVM Radial and Random Forest text-based machine learning models) outperforms both the Volatility (volatility-based) and Sentiment (sentiment score-based) Models, with an area under curve of 0.9105. The Averaged Model (averaged across the Average Text Model, Volatility Model, and Sentiment Model) performs best, with an area under the curve of 0.9165.

We can further look at the model performance for each country based on a true positive rate threshold of 55 percent. This ensures that we are able to identify over half of the financial crises in our sample without having too many false positives. For the Averaged Model and the Average Text Model, for example, the false positive rate is smaller than 5 percent. As shown in the confusion matrix in Figure 8, it can be easily seen that the Volatility, Sentiment, and Average Text Models all convey slightly different information in identifying minor or more severe crisis based on false positives or false negatives given a 55 percent true positive rate. Here, "FN", "FP", "TN", and "TP" stand for false negative, false positive, true negative, and true positive observations, respectively. Therefore, the lighter colors in the confusion matrix indicate that better out-of-sample performance was

recorded for that model. Whereas the Average Text Model, Volatility Model, and the Sentiment Model all provide many false positives and false negatives individually for many countries, the Averaged Model perfectly identifies periods of financial crises and periods without financial crises in countries such as Turkey, the Netherlands, Finland, Denmark, Canada, and Belgium, and the model output from the Average Text Model predicts crisis/non-crisis periods without error for countries such as the Netherlands, Canada, Belgium, and Australia.

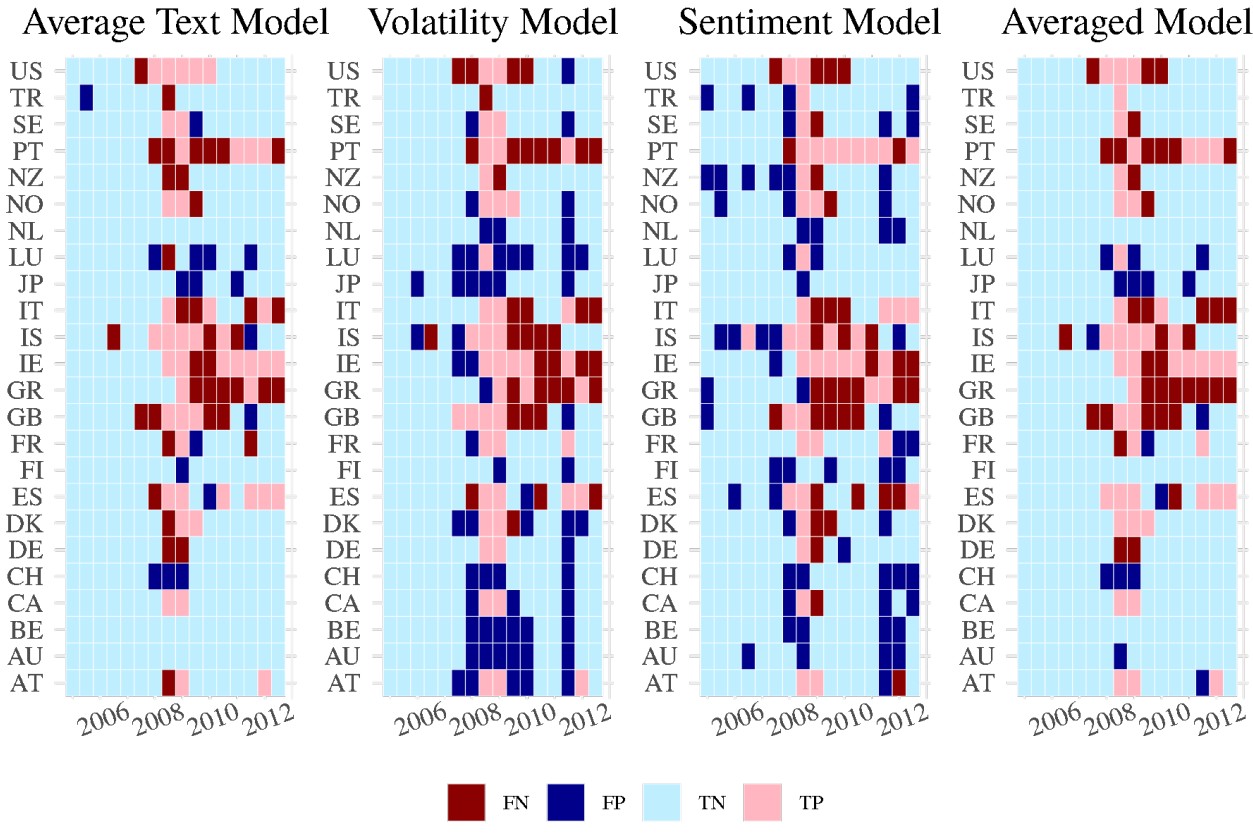

**Figure 8.** **Confusion Matrix for R&R Minor Crises on OECD Text**. *Note:* This figure shows confusion matrices for the out-of-sample period of 2005–2012 at the true positive rate (TPR) threshold of 55 percent for models predicting Romer and Romer (2017) 5+ severity crises trained on OECD text from 1981–2004. The confusion matrices indicate whether the model crisis classification for each country and time period (half year) is a False Negative (FN), False Positive (FP), True Negative (TN), or True Positive (TP). The lighter colors in the confusion matrix indicate better out-of-sample performance. The results show that the Average Text Model and Averaged Model outperform the Volatility and Sentiment Models. The Average Text Model performs perfectly for the Netherlands, Canada, Belgium, and Australia, while the Averaged Model performs perfectly for Turkey, the Netherlands, Finland, Denmark, Canada, and Belgium. Country abbreviations are as follows: United States (US), Turkey (TR), Sweden (SE), Portugal (PT), New Zealand (NZ), Norway (NO), Netherlands (NL), Luxembourg (LU), Japan (JP), Italy (IT), Iceland (IS), Ireland (IE), Greece (GR), United Kingdom (GB), France (FR), Finland (FI), Spain (ES), Denmark (DK), Germany (DE), Switzerland (CH), Canada (CA), Belgium (BE), Australia (AU), Austria (AT).

For a completely out-of-sample experiment, we can look at the model results for a country that is not in the training sample, which in our case is Mexico. Mexico joined the OECD much later than the 24 countries in our sample, meaning that its OECD Economics Outlooks start in the mid-1990s. As such, they are excluded from the Romer and Romer (2017) crises dataset and from our main training sample. As shown in Figure 9, on OECD text the Average Text Model indicates with more than 60 percent probability that Mexico

was in a crisis during the mid-1990s, more widely known as the "Tequila Crisis". Mexico suffered through a banking crisis from 1994 to 1996 according to Laeven and Valencia (2013), and a currency crisis from 1994 to 1995 according to Reinhart and Rogoff (2009).[4] In addition, it can be seen that the height of COVID-19 pandemic period was associated with a crisis, though with a somewhat lower probability.

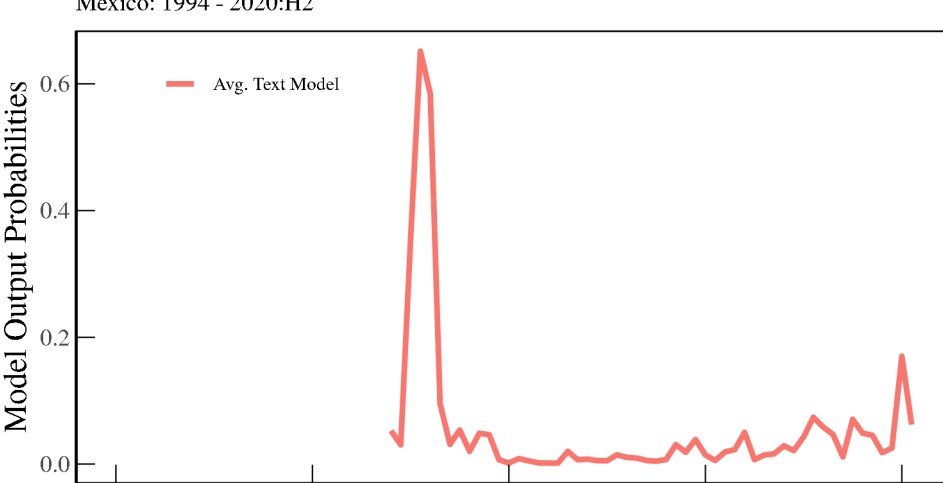

**Figure 9. Average Text Model Output for Mexico**. *Note:* This figure demonstrates a completely out-of-sample experiment on Mexico's OECD text using the Average Text model trained on 1981–2004 OECD text for Romer 5+. Mexico is not in the training set, as its OECD Economic Outlooks did not start until the mid-1990s; thus, it is excluded from the Romer and Romer (2017) crises dataset and from our main training sample. This chart shows that the Average Text model is able to identify the 1994 Mexico Peso Crisis as an elevated probability of crisis in the mid–1990s.

In order to determine whether our text-based models work as intended, we can look more closely at how individual words or bigrams affect the probability of the SVM Radial model based on Local Interpretable Model-agnostic Explanation (LIME) based on Ribeiro et al. (2016), which identifies an interpretable model over the interpretable representation that is locally faithful to the classifier model. These types of models are used in the literature to ensure interpretability of model outputs. In Figure 10, it is noticeable that the top 30 factors in terms of LIME feature weights include many of the important variables listed in Figure 6. Here, however, we can see how much they contribute to the model output probability on average. For example, a mention of "bank" in an OECD Economic Outlook text contributes to an increase of approximately 7 percentage points on average in the model output's probability of a country currently being classified as in a minor or severe crisis. In contrast, words such as "strong" are on average associated with a decrease in this probability.

LIME provides a method of unveiling the black-box nature of machine learning algorithms by running a model with and without the words of interest on specific text, even locally. To provide a more specific example, Figure 11 shows an example o the OECD Economic Outlook text for the first half of 2008 in the United States, with red words contributing to a higher model output probability that the United States in a Romer 5+ crisis and blue words contributing to a lower probability (the darker the shading of a word, the higher the magnitude of the contribution). Again, words such as "bank" and "credit" are a dark shade of red, implying that these terms help to signal a crisis. This is consistent with the fact, widely acknowledged in the literature, that the 2008 crisis was a banking-driven crisis.

## Lime Weights, distribution across top features

**Figure 10.** **Top 30 Average Feature Weights for Minor Crises on OECD Text**. *Note:* Local Interpretable Model-agnostic Explanations (LIME) provides a method of unveiling the black-box nature of machine learning algorithms by running a model with and without features (terms). This figure shows how much, on average, the top 30 terms according to LIME feature weights contribute to the model probability outputs of crisis. For example, the mention of "bank" in the OECD Economic Outlook text contributes mostly 0.5 to 1 percentage point to the model probability output of an observation being in a minor or more severe crisis. In contrast, words such as "strong" are associated with a decrease in this model probability output on average. These results are based on the SVM Radial model trained on OECD text from 1981–2004 predicting Romer 5+ crises.

Considering the context of the OECD Economic Outlook Reports, these are intuitive results. Documents inundated with references to the banking sector, loans, and other financial developments are quite unusual compared with normal circumstances, and convey that the financial sector is weighing on real economic activity. This is exactly what Romer and Romer (2017) tries to identify, that is, whether problems in the financial sector are having real effects.

US 2008:H1
The US economy is at the epicentre of a financial crisis , which is causing considerable disruption to real activity . The trigger for the crisis was a sharp rise in delinquencies on subprime mortgages , which led to large losses on the securities backed by these mortgages . As investors came to realise that mortgage - and asset - based securities were much riskier than supposed , demand for and trading of such products dried up , resulting in further losses on a variety of credit - based securities . Banking institutions linked to these leveraged products incurred large losses , necessitating measures to restore their financial health . This involves a Banks have tightened lending standards [1]
The housing market is tumbling...

**Figure 11.** **US OECD Economic Outlook with LIME Highlighting for 2008**. *Note:* This passage shows the OECD Economic Outlook text for the first half of 2008 in the United States shaded according to LIME feature weights. Red words indicate a positive contribution to model output probability of the United States being a Romer 5+ crisis, while blue words indicate a negative contribution; the darker the shading of a word, the higher the magnitude of the contribution. In other words, dark red words contribute more (and dark blue contribute less) to the probability of the machine learning algorithm's determination that the United States is in a financial crisis. Underlined terms are those in the OECD dictionary. The word "bank" is quite a dark shade of red, implying that this term helps to signal that the United States is in a crisis. These results are based on the SVM Radial model trained on OECD text from 1981–2004 predicting Romer 5+ crises.

To provide an example of how this can be useful for detecting new types of financial crises during highly unusual economic periods, we can look at the Average Text Model for the United States up to 2020 and at our LIME results during the height of the COVID-19 pandemic health crisis to see whether our model determines that these were financial crises. As shown in Figure 12, the model does well at detecting the Savings and Loan Crisis in 1990 in-sample and the Global Financial Crisis around 2008 out-of-sample (after the dashed vertical line) according to Romer and Romer (2017). When we look at the 2020 period during the COVID-19 pandemic, the model picks up unusual wording in the OECD Economic Outlooks for the United States during this period, with the crisis probability spiking to close to 50 percent in the first half of 2020.

Textual Methods: Nowcasting

United States: 1981 - 2020:H2

Shaded areas show Romer Crises of severity 5 or higher

**Figure 12.** **Average Text Model Output for United States**. *Note:* This figure shows nowcasting results of model probability for a Romer and Romer (2017) 5+ severity crisis in the US (model trained on OECD text from 1981–2004). Periods indiacted as a Romer and Romer 5+ crisis are shaded in pink. The figure shows a spike in the probability of a crisis in the first half of 2020. Because the Romer and Romer crises dataset ends in 2012, the post-2012 predictions here are completely out-of-sample.

If you look at the LIME results for this period, Figure 13 shows that, unlike in the 2008 crisis period, words and terms such as "government", "financial difficulties", "financial support", and "unemployment" appear to be driving the model to indicate that a financial crisis is occurring, as opposed to the banking and credit-related words and terms encountered in 2008. This shows how text-based models can help to detect different types of financial crises even in a highly unusual economic environment such as the COVID-19 pandemic period.

US 2020:H1
United States
The COVID - 19 outbreak has brought the longest economic expansion on record to a juddering halt . GDP contracted by 5 % in the first quarter at an annualised rate , and the unemployment rate has risen precipitously . If there is another virus outbreak later in the year , GDP is expected to fall by over 8 % in 2020 ( the double - hit scenario ) . If , on the other hand , the virus outbreak subsides by the summer and further lockdowns are avoided ( the single - hit scenario ) , the impact on annual growth is estimated to be a percentage point less . The unemployment rate will remain elevated after states lift their shelter - in - place orders , reflecting ongoing difficulties in sectors such as hospitality and transportation , and the sheer scale of job losses . With unemployment remaining high , inflation is projected to stay low , although less so if subsequent lockdowns are avoided .
Massive monetary and fiscal responses have shielded households and businesses , but more will be needed to reduce lingering effects such as large numbers of bankruptcies and labour - market exits . Complementary payments to augment unemployment insurance should continue , while the tax burden of households and businesses should be lowered when they are directly affected by the lockdown . Additional support will be needed to help workers return to work . Some states and local governments will face financial difficulties as their main revenue sources have dried up , and their debt burden will need to be addressed . Importantly , well - designed public financial support for developing a vaccine and treatment of COVID - 19 could help prevent a recurrence of a pandemic again leading to deaths and debilitating the economy .

**Figure 13. US OECD Economic Outlook with LIME Highlighting for 2020**. *Note:* This passage shows the OECD Economic Outlook text for the first half of 2020 in the United States shaded according to LIME feature weights. The red words indicate a positive contribution to model output probability that the United States in a Romer 5+ crisis, while blue words indicate a negative contribution; darker shading of a word indicates a higher magnitude of contribution. In other words, darker red words contribute more (and darker blue contribute less) to the probability of the machine learning algorithm indicating that the United States is in a financial crisis. Underlined terms are those in the OECD dictionary. The words "government", "financial", and "unemployment" are shaded in red, implying that these terms help to signal that the United States is in a crisis. These results are based on the SVM Radial model trained on OECD text from 1981–2004 predicting Romer 5+ crises.

Up to now, we have shown results from models based on minor crises with severity of 5 or more. However, we can run our machine learning and other models on different levels of crisis severity. In fact, as in Figure 14, we find that as we vary the degree of crises to be identified (starting from minor credit disruptions to more severe crises), the Averaged Model generally outperforms the other models; moreover, it consistently outperforms the Volatility Model, especially when the severity of the crisis increases.[5] The area under the ROC curve peaks at around 0.90 when attempting out-of-sample nowcasting for Romer and Romer (2017) crisis severities of 5 or more. As the level of crisis to be identified increases in its severity, the information content purely from the OECD Economic Outlook text outperforms even the Averaged Model, as the signal-to-noise ratio deteriorates rapidly for the Volatility Model. Indeed, as mentioned earlier, volatility in financial markets provides many false positives when it comes to detecting financial crises. A sharp increase in volatility in equity markets does not necessarily mean that a financial crisis is occurring. In particular, it provides a noisy signal when it comes to detecting the severest of crises. Although sentiment appears to help detecting crises as they become more severe, it is far inferior to the information content embedded in the actual words and terms in the OECD Economic Outlooks.

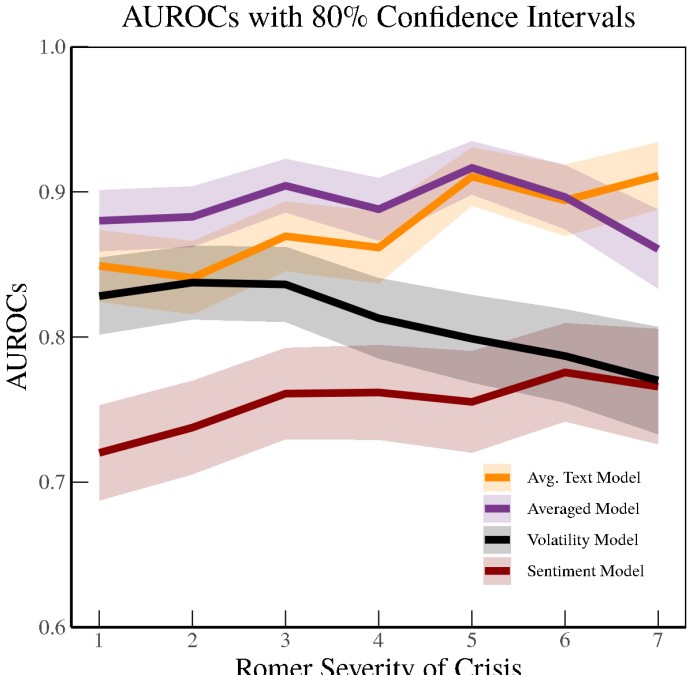

**Figure 14. Results for Various R&R Crises on OECD Text**. *Note:* This figure shows the Area Under ROC (AUC) for the Average Text Model, Averaged Model, Volatility Model, and Sentiment Model with bands of 80% confidence intervals across Romer and Romer (2017) severity 1+ through 7+. When varying the degree of crisis to be identified (starting from minor credit disruptions to more severe crises), the Averaged Model generally outperforms the other models, and consistently outperforms the Volatility and Sentiment Models. The area under the ROC curve peaks about at 0.90 when attempting out-of-sample nowcasting of Romer and Romer (2017) crisis severities of 5 or more. As the severity of the crisis to be identified increases, the information content in the OECD Economic Outlook text eventually outperforms even the Average Model, as the signal-to-noise ratio deteriorates rapidly for the Volatility Model. All results here are based on models trained on OECD text from 1981–2004.

### 5.2. Identifying Laeven and Valencia Crises using RDNF Text and Combinations of Models

Next, we implement the same models using the media-based Refinitiv RDNF articles. Figure 15 shows the results from a similar exercise that takes advantage of a wider cross-section of countries in two datasets, namely, Laeven and Valencia (2013) crisis data, which are considered most closely related to "minor" or "moderate" crises in Romer and Romer (2017), and the RDNF data, which have more than double the cross-section of countries of the OECD data, in addition to a higher monthly frequency. Here, the Volatility Model and the Averaged Model work best, the Sentiment Model is next best, and the Average Text Model underperforms, with AUCs of about 0.80, 0.75, and 0.62, respectively. Volatility appears to be important in detecting crises when using a larger set of countries that includes more emerging markets than the OECD sample. As in the literature, it is interesting to see the high performance of the Sentiment Model, as it represents a good indicator for summarizing what terms are being mentioned in the media.

The IMF Article IVs have a broader set of countries in the sample, as in the RDNF; however, they are not published regularly, and have many gaps in between years. Figure 16 shows the AUCs for the different models using IMF Article IV text and Laeven and Valencia (2013) banking crises; similarly to the models based on the RDNF, it can be seen that the Volatility Model works best, followed by the Average Model, with AUCs of 0.89 and 0.88, respectively. Again, volatility is a good measure when thinking about crises in a wider set of countries. The Sentiment Model does not perform as well as on the RDNF data.

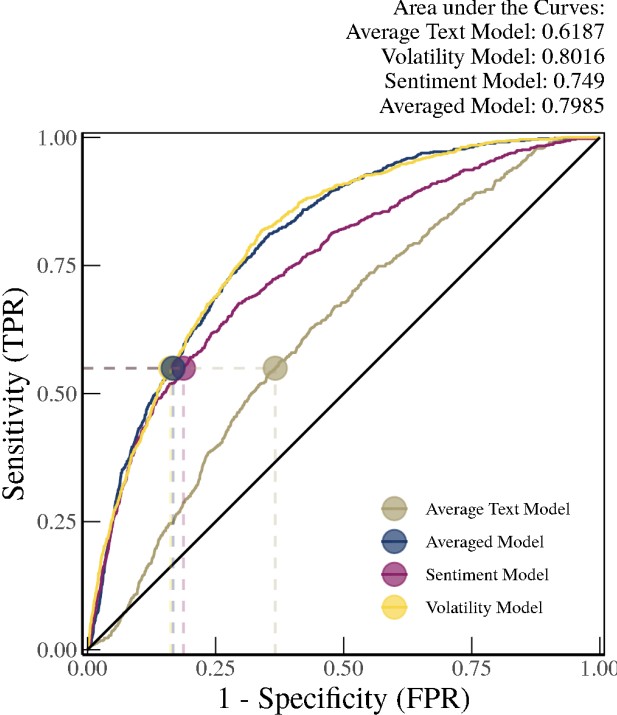

**Figure 15. Results for LV Banking Crises on RDNF Text**. *Note:* This figure shows the ROC curves for our out-of-sample period (2005-2017) for models trained on the period 1996–2004 using RDNF text based on Laeven and Valencia (2013) banking crises. The results show that the Volatility Model performs best (0.8016), followed by the Averaged Model, Sentiment Model, and Average Text Model.

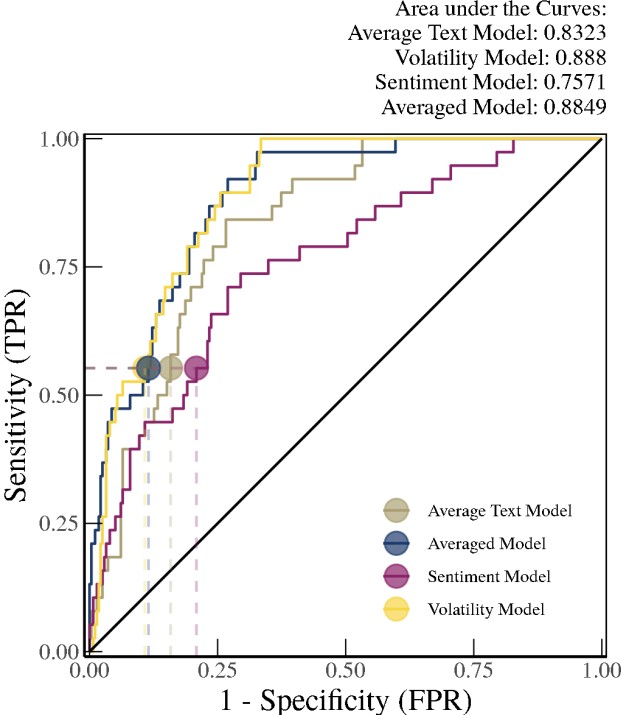

**Figure 16. Results for LV Banking Crises on IMF Article IVs**. *Note:* This figure shows the ROC curves for our out-of-sample period (2005–2017) for models trained on the period 1983–2004 using IMF Article IV text based on Laeven and Valencia (2013) banking crises. The results show that the Volatility Model performs best (AUC of 0.888), followed by the Averaged Model, the Average Text Model, and the Sentiment Model.

We can use different models based on different texts to enhance our out-of-sample forecasting. As a template, we can then choose the models that work relatively well and combine them. In our next example, we introduce a combination model that combines the model outputs of the Average Text Model based on Romer and Romer (2017) crises trained on OECD Economic Outlooks, the Volatility Model and Sentiment Model based on Laeven and Valencia (2013) crises trained on RDNF data, and the Average Text Model based on Laeven and Valencia (2013) crises trained on IMF Article IVs. Figure 17 illustrates the results, showing that the area under the ROC curve can be pushed up to 0.92 with this combination model. Although this is not a statistically significantly difference from the Average Text Model based on Romer and Romer (2017) trained on OECD text, it is more than a percentage point higher, and could be potentially more meaningful with additional optimal weighting of the different models.

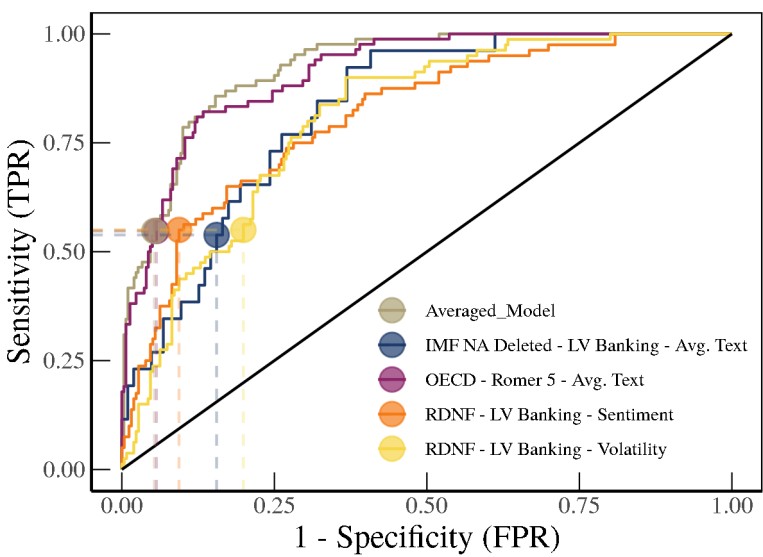

**Figure 17. Results for Combined Model.** *Note:* This figure shows the ROC curves for our out-of-sample period (post-2005). We introduce a combination model by combining the model outputs of the models that work well: the Average Text Model predicting moderate crises according to Romer and Romer (2017) using OECD Economic Outlook text, the Volatility and Sentiment Models predicting banking crises according to Laeven and Valencia (2013) using RDNF text, and the Average Text Model predicting banking crises using IMF Article IV text. The figure shows that averaging the model predictions of these four models (referred to as the "Averaged Model" in the figure) boosts the area under the curve (AUC) to 0.9225. All models were trained on pre-2005 data.

Finally, to check whether our Chen–DeHaven–Kitschelt–Lee–Sicilian (CDKLS) model (the Averaged Model trained on the period 1981–2004 using OECD text predicting Romer crises 5+) can meaningfully identify crisis periods, we use out-of-sample crisis identifications to conduct local projections, as in Romer and Romer (2017). The lower left panel in Figure 18 shows that our model provides similar magnitudes in terms of the impulse response to crisis periods when it comes to cumulative effects on GDP growth as compared to using other crisis definitions (Reinhart and Rogoff (2009), Romer and Romer (2017), or IMF or Laeven and Valencia (2013)) for the 24 OECD countries in our sample. In particular, as compared to Laeven and Valencia (2013) crises, the bands around the impulse responses

are significantly smaller, and are comparable to crises in other studies. This shows that our definition provides a clear indication that it is picking up financial disturbances that affect the real economy in a meaningful way, as in the crises identified by other studies, which is at the core of how experts define financial crises. Unlike these other definitions of crises, however, our CDKLS index can be updated in real time without significant lag, and can to providing a consistent structure able to perform identification across countries without as much judgment bias as exists in other methods.

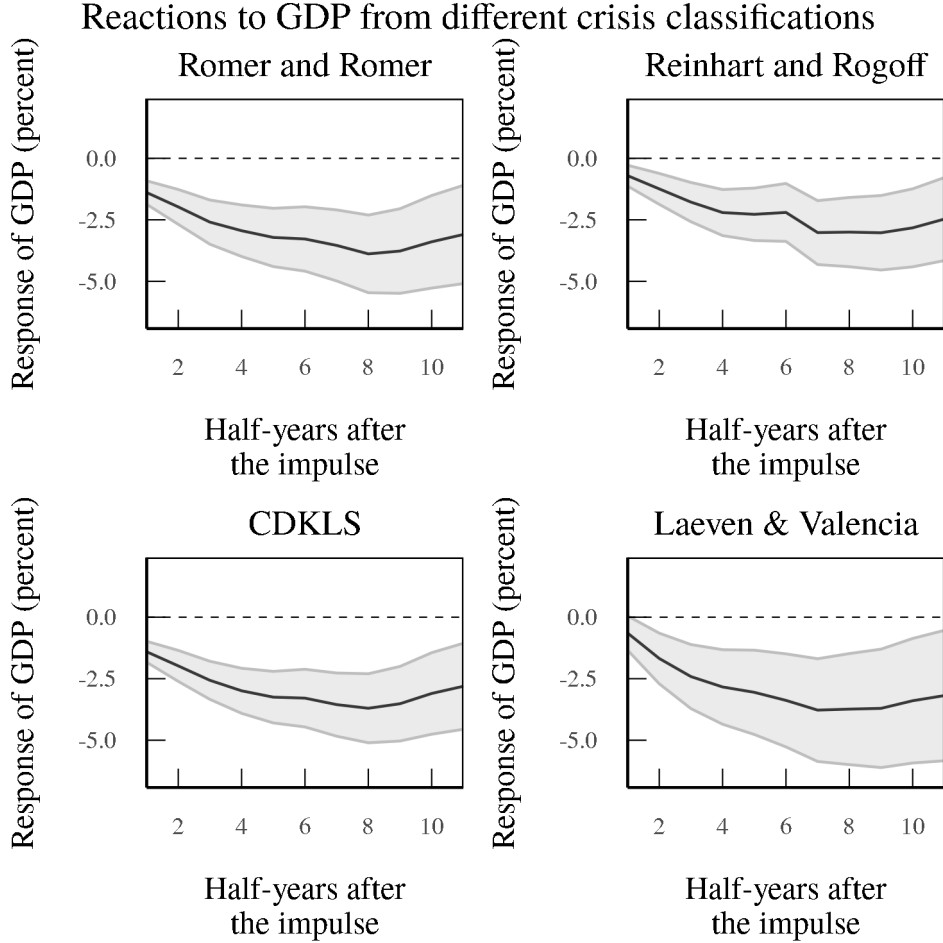

**Figure 18. Local Projection Results**. *Note:* This figure shows the predictions of Romer 5+ crises on 2005–2012 OECD text at the 0.55 true positive rate for the Chen–DeHaven–Kitschelt–Lee–Sicilian (CDKLS) model (**bottom left**) and the Average Model trained on the period 1981–2004 using OECD text. It can be seen that both models have similar magnitudes in terms of the impulse response of GDP compared to expert-identified crises: Romer and Romer (2017) (**top left**), Reinhart and Rogoff (2009) (**top right**), and Laeven and Valencia (2013) (**bottom right**).

## 6. Forecasting and Backcasting

In this section, we undertake forecasting and backcasting exercises to see which text helps more to shed light on crises in the future or the past, respectively. Again, we trained a model with data up to 2004 and show out-of-sample results for forecasting and backcasting minor (Romer and Romer (2017) severity 5 or more) Romer and Romer (2017) crises using OECD Economic Outlook text and comparatively, using RDNF text. In addition, we look at expanding models by taking on new incoming data instead of stopping in 2004. For all models, we forecast crises *x* periods ahead or behind. As Figures 19 and 20 indicate, the OECD SVM Radial Model is quite dominant in backcasting as far back as two years, and keeps its edge into the nowcasting period whether using the chunk method or the expanding method. However, when forecasting even six months ahead, the models based

on OECD Economic Outlook text face a steep drop-off, while the models based on RDNF data remain relatively elevated compared to backcasting. When forecasting one year ahead, the OECD model has an AUC of barely 0.50.

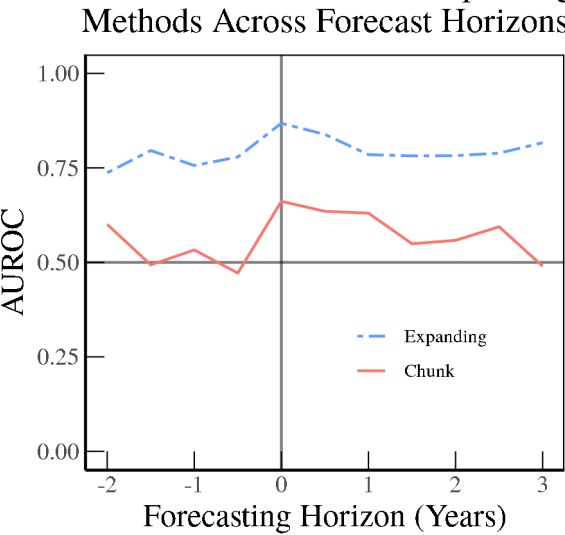

**Figure 19. Results for R&R Minor Crises on OECD Text**. *Note:* This figure shows the area under the ROC curve for the SVM Radial Model based on OECD text and Romer and Romer (2017) 5+ severity for backcast and forecast horizons of −2 to 3 years (at half-year frequency) for the "chunk" for longer time horizons. The "expanding" method shows a more gradual decrease in model performance.

**Figure 20. Results for R&R Minor Crises on RDNF Text**. *Note:* This figure shows the area under ROC curve for models based on RDNF text and Romer and Romer (2017) banking crises for backcasting and forecasting horizons of −2 to 3 years (at a half-year frequency) for the "chunk" and "expanding methods". The results show that the nowcasting (horizon = 0 years) model performs best, with a decline in performance at longer time horizons, while the "expanding" method shows a more gradual decrease in model performance at longer time horizons.

In contrast to the SVM Radial Model using official OECD text, the SVM Radial Model based on the media (RDNF) perform relatively better in forecasting than in backcasting, as indicated in Figure 20 as compared to the results for the OECD text. Then using the chunk

method, the model based on RDNF holds up even one year into the future, then begins to gradually decline.

Here, we illustrate how retraining on an expanding sample helps in terms of forecasting. In particular, compared to the OECD text (Figure 19), the RDNF SVM Radial Model expanding model greatly outperforms the chunk model even up to a three-year horizon (Figure 20) without losing much of its predictive power. This illustrates that an abundance of text can help in learning new relationships when the sample is augmented for training machine learning models.

Textual data can potentially aid in the identification of financial events in two ways: through their descriptive properties, or through their instigative potential. In order for either explanation to be valid, researchers must be able to extract information from text in a consistent and automated fashion.

The descriptive explanation is that the authors of a text convey information about the financial landscape that is relevant to the likelihood of events of interest coming to pass. The authors may, for example, use more positive words in good times and more negative words in bad times. An important element of this mechanism by which text can help identify financial events is that the circulation of the text is less relevant; it makes little difference if the text is read by millions or by no one. This explanation follows the literature in suggesting that text contains useful information and is an important source of information for understanding economic phenomena (Gentzkow et al. (2019)).

Alternatively, text can add predictive power when identifying financial events by influencing whether such events happen. This is the instigative, or at least partially causal, mechanism, which goes much further than being simply descriptive and an additional source of information. For example, Shiller (2017) emphasizes that market participants and economic agents can be driven by a narrative, which in turn can be driven by what is written in text. It is not necessary that the text contain any descriptive truth at the time of its publication as long as the text changes the financial landscape in a predictable way. A classic example of this mechanism is a bank run; regardless of whether a particular community bank has a cash shortage, an article stating as much published by an influential local paper can become a self-fulfilling prophecy. Note that the effect of such an article is likely to diminish with a smaller readership (Iyer et al. (2016)).

It is likely that most financially-relevant texts operate to identify financial events in part through both of these mechanisms. OECD Economic Outlook Reports (OECD) have a much lower readership than Refinitiv RDNF articles; thus, one might expect TRNA to be relatively more instigative. On the other hand, readers of OECD Reports might have more insight into the financial landscape or into policies which affect the landscape. Disentangling these mechanisms is difficult, especially with no reliable readership data; nonetheless, it is possible to draw conclusions about which mechanisms are more impactful from different text sources, as we attempt in the following results and discussion sections.

Our results on trying to backcast, nowcast, and forecast financial crises using different types of text provide food for thought in this debate. Our findings are consistent with the view that most financially relevant texts operate to identify financial events in part through both of these mechanisms. OECD Economic Outlook Reports (OECD) have a much lower readership than RDNF articles, and as such one might expect RDNF to be relatively more instigative. On the other hand, readers of OECD Reports might have more insight into the financial landscape or into policies which affect the landscape. These differences in informational content may drive our results, in that OECD Economic Outlooks may be better at historical description and current insights into how the financial landscape is changing, while RDNF articles may provide a more forward bent in describing financial developments.

Finally, using only OECD Economic Outlook data, we can look at how the models work when trying to backcast/nowcast/forecast the exact onset of a crisis by not including crisis periods except for the initial period leading up to it when training a variety of models. For example, we can include the credit-to-GDP gap as an indicator in addition to the Sentiment

Model, SVM Model, Volatility Model, and Averaged Model of all four models. Figure 21 illustrates that the Sentiment, SVM, Volatility, and Averaged Model all perform better at nowcasting the onset of crises compared to backcasting and forecasting, and struggle the most in forecasting the exact timing of the onset of a crisis, especially at a distance of one to two years out. On the other hand, the Credit Models perform relatively better in forecasting crises one or two years out, whether using the chunk method or the expanding method, with an AUC of between 0.60 and 0.80 compared to other models. The Credit Models perform the best out of all the models at forecasting the onset of crises two years out. One possible explanation for these models performing better at this forecasting horizon is that credit-to-GDP is more predictive at longer horizons, while models that include OECD text or volatility capture more contemporaneous or backward-looking information. The OECD text may especially capture more backward-looking information, with the text SVM model performing the best at backcasting the onset of crises.

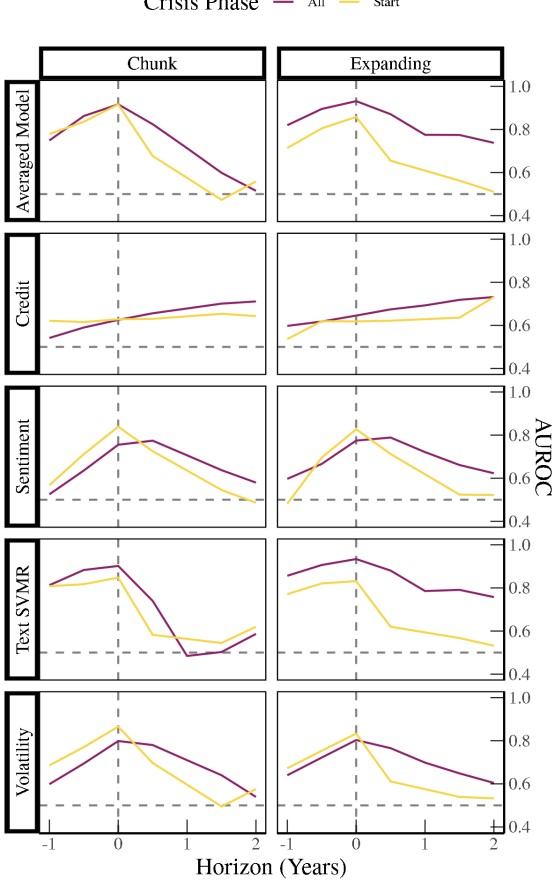

**Figure 21. Crisis Phase: Starts vs. All**. *Note:* This figure shows area under the ROC curve (AUROC) for models based on OECD text and Romer and Romer (2017) 5+ severity for backcasting and forecasting horizons of −1 to 2 years (at half-year frequency) for the "chunk" and "expanding" methods for the full set of crisis phases (i.e., "All") and observations that indicate the onset of crisis phases (i.e., "Start"). For the most part, all models are better at predicting "All" crisis phases versus "Start" crisis phases. The Sentiment, SVM, Volatility, and Averaged Model (average of SVM Radial, Sentiment, and Volatility Models) all perform better at nowcasting the onset of crises compared to backcasting and forecasting, and struggle the most in forecasting the exact timing of the onset of a crisis, especially from one to two years out. On the other hand, the Credit Models perform relatively better at forecasting crises one to two years out, whether using the chunk method or the expanding method, with an AUC of between 0.60 and 0.80 compared to other models; moreover, they perform the best out of all the models at forecasting the onset of crises two years out.

## 7. Conclusions

Using machine learning, we have found that text can help to identify and predict financial crises in real time without having to wait on sometimes inconsistent expert judgment in lagged time. In particular, different types of text can be trained on different crisis definitions to better identify differences in crisis severity. As expected, the OECD textual models perform quite well at identifying Romer and Romer (2017) crises. The RDNF sentiment seems to provide a lot of information as well, with the added benefit of access to a broader set of countries. While IMF Article IVs have inconsistent frequency, they clearly add information due to their relevance in writing about financial stress or financial crises. A naive combination of sources shows potential improvements over individual text sources. Finally, text data appear to provide statistically significant improvement over a baseline model with just volatility, especially for detecting more severe types of crises.

Our results may shed light on the channels by which OECD and RDNF text models can identify financial crises. Recalling the two mechanisms outlined earlier, namely, descriptive and instigative, it is natural to think that the descriptive mechanism would be more powerful for backcasting than for forecasting. This is because it is easier to describe the financial landscape of the present or past than it is to describe the financial landscape of the future. This in turn leads us to expect that texts with a relative advantage in backcasting may operate more strongly through the descriptive mechanism. We bolster this expectation with the observation that the instigative mechanism cannot apply to backcasting at all, as it is impossible for a text to instigate action in the past.

Taken together, and combined with our results indicating that RDNF performs relatively better in forecasting than backcasting and that the opposite is true of OECD, these dynamics point towards the conclusion that written news, or at least RDNF text, is relatively more instigative than descriptive compared to OECD outlook text. Of course, there may be other explanations. Two text sources operating solely though the descriptive mechanism might differ in forecasting/backcasting power simply because one attempts to describe the past and the other attempts to describe the future. While it is possible that RDNF generally attempts to describe the future while OECD attempts to describe the past, we think that this is unlikely, as the OECD reports are explicitly meant to be "outlooks", whereas RDNF focuses almost entirely on real-time description of events in the present.

Understanding how text can help in understanding financial crises has a number of implications. Various machine learning and econometric methods can potentially be applied to a variety of types of crises using a range of different text sources, which can help policymakers to determine where a particular financial system is in the financial cycle, or more specifically whether the financial system is weighing on real economic activity in a consistent manner. In turn, this can be useful for macroprudential policy, monetary policy, and even fiscal policy, as different phases in the financial cycle have different implications for real economic activity. Moreover, building a framework that is consistent across countries in real time can benefit policymakers around the world, especially when international coordination is required across different government policies.

**Author Contributions:** Conceptualization, M.C., M.D., S.J.L., and M.J.S.; methodology, M.C., M.D., S.J.L., and M.J.S.; software, M.C., M.D., I.K., and M.J.S.; validation, M.C., M.D., I.K., S.J.L., and M.J.S.; formal analysis, M.C., M.D., I.K., and M.J.S.; investigation, M.C., M.D., I.K., S.J.L., and M.J.S.; resources, M.C., M.D., I.K., S.J.L., and M.J.S.; data curation, M.C., M.D., I.K., and M.J.S.; writing—original draft preparation, S.J.L.; writing—review and editing, M.C., M.D., I.K., S.J.L., and M.J.S.; visualization, M.C., M.D., I.K., and M.J.S.; supervision, S.J.L.; project administration, S.J.L.; funding acquisition, S.J.L. All authors have read and agreed to the published version of the manuscript.

**Funding:** This research received no external funding.

**Data Availability Statement:** Publicly available data will be provided upon reasonable request.

**Acknowledgments:** We thank three anonymous referees for their feedback. We are particularly grateful to Arthur Turrell for guiding us through the initial phases of our research and Leo Saenger for excellent research assistance. We thank Xiang Li and Robin Lumsdaine for terrific discussions of

the paper and Ricardo Correa for helpful comments on the paper. We also thank seminar participants Federal Reserve Board, Federal Reserve Bank of Richmond, and the ECB/ESRB, and conference participants at the SEM Conference, IBEFA Meeting, International Conference on Advanced Research Methods and Analytics, RiskLab/Bank of Finland/ESRB Conference on Systemic Risk Analytics, SFA Meeting, FMA Meeting, NFA Conference, EEA Congress, AEA Meeting, Joint FRB-IMF Workshop on New Techniques and Data in Macro Finance, the Textual Analysis in Economics and Finance Research Conference, the 3rd Forecasting at Central Banks Conference at the Bank of Canada, and the 3rd Conference on Financial Stability at the Bank of Mexico for valuable feedback on our analysis. The views stated herein are those of the authors and are not necessarily the views of the Federal Reserve Board or the Federal Reserve Bank of Boston. All errors are the authors' and no one else's.

**Conflicts of Interest:** The authors declare no conflict of interest.

## Appendix A. Data Cleaning

*Appendix A.1. OECD Text Data*

OECD Economic Outlooks are downloaded as PDFs from the OECD iLibrary, which provides all of the documents going back to 1967. The PDFs are converted into HTML files, which then allows blocks of text, font size, and the position of the text on the page to be identified. This allows the document to be scanned in order to identify the sections that are about specific countries; these almost always start with a header line for that country, i.e., "CANADA", in large font.

After assigning all of the scanned documents to specific countries, we removed numbers, stop words ("the", "and", etc.), and short words (less than three characters), converted all text to lower case, and lemmatized the remaining words using the R package *textstem* (Rinker 2018b) and the *lexicon* (Rinker 2018a) engine. This method combines various versions of a word (e.g., "fall", "fell", and "falling") into one instance of that word ("fall"). This is preferable to stemming, which removes common endings to words and can miss certain peculiarities of the English language.

Finally, we created bigrams from the remaining text. This process joins words that are adjacent, for instance, "federal" and "reserve" becomes "federal_reserve"; the original unigrams were kept along with the bigrams. One additional cleaning of the text involved differentiating between "bank" used in the context of commercial banks as opposed to "bank" when used in the context of central banks by creating a list of central banks and substituting the proper noun of such banks into just a generic "central bank." For example, "Bank of Japan" would be transformed into "central bank' in our data.

*Appendix A.2. Refinitiv RDNF Text Data*

We extensively cleaned and sorted articles from the Refinitiv RDNF, which contains around 66.3 million articles. After dropping all articles written in languages other than English, we were left with 42.5 million articles. We constructed a story chain identifier, which sequentially links articles that are updates to original articles. We only kept the first article in a story chain, and discarded any articles that were updates, leaving 28.1 million articles. Finally, we dropped articles classified as "repeated works", (i.e., obituaries, weekly oil readouts), alerts, or headlines with no body text, for a final total of 19.3 million articles.

Because the Archive does not identify the country or countries about which articles are written, we used an algorithm to classify articles by country. Using the R package *newsmap* (Watanabe 2018), we first used a seed dictionary of proper nouns that map to countries to make a first pass over the corpus in order to find other proper nouns to add to the dictionary. All articles were then classified again in a second round using the expanded dictionary. The *newsmap* (Watanabe 2018) package assigns each article a score for each country, with higher scores meaning that the article is more likely to be about a certain country. Articles may have high scores for multiple countries. As an illustration of the process, the seed dictionary associates "Paris", "France", and "French" with the country "France". If any of these words are found in an article, the algorithm assigns the article a high score for the "France" category. Then, *newsmap* (Watanabe 2018) finds other words that are common in these articles that are

most likely to be classified as "France", such as "Sorbonne", and assigns articles that contain the word "Sorbonne" a high score in the"France" category.

We manually classified a random sample of 560 articles and compared these to the scores output by the algorithm. Using a score cutoff of 2.25 for the highest-value country for each article and a score cutoff of 3.5 for the 2nd–5th countries resulted in a 90% confidence rating for the average classification. That is, a random country–article classification in the results is 90% likely to be correct, with 'correct' being defined by manual verification. In the end, we used a cutoff of 0.0 for the first country, 2.5 for the second and third, and discarded the rest. After this process was complete, 87% of the articles were classified with at least one country, 3.6% of which were classified with two countries and roughly 1% of which were classified with three countries, with 82% confidence. Using these results, 238 unique countries were contained in the RDNF text.

The RDNF articles were then subset to include only articles containing at least one relevant economics-related word. These words were "financ-", "stability", "econ-", "market", "investment", "trade", "stock", "sovereign", "debt", "bank", and "assets". Articles that contained at least one sports-related word were removed. These words were "soccer", "championship", "World Cup", "basketball", "football", "baseball", "sport", "tennis", and "Olympic".

Articles were aggregated to a monthly frequency. We filtered to only country-months with at least fifty articles and only countries with at least 75 percent of months meeting this condition.

We removed numbers, stop words ("the", "and", etc.), and short words (less than three characters), and converted all text to lower case. We then lemmatized the remaining words using the R package *textstem* (Rinker 2018b) and the *lexicon* (Rinker 2018a) engine. This process combines various versions of a word (e.g., "fall", "fell", and "falling") into one instance of that word ("fall"). This is preferable to stemming, which removes common endings to words, as it can miss certain peculiarities of the English language.

The process for creating bigrams and transforming proper central bank names to "central bank" is the same as for the OECD Economic Outlooks.

*Appendix A.3. IMF Article IV Text Data*

Staff Reports for Article IV Consultations are downloaded as PDFs from the Archives Catalog from the International Monetary Fund. While there is at most one Article IV Consultation per country per year, there may be multiple iterations of Staff Reports available for download for the same Article IV Consultation. In these cases, we kept only the most complete Staff Report at the latest release date. In such cases, we use this release date as the date of publication instead of the IMF's stated year for the Article IV Consultation in our analysis.

We used Adobe Acrobat's OCR to convert scanned PDFs to searchable text PDFs. These PDFs were converted into HTML files, from which we extracted only the text. We removed numbers, stop words ("the", "and", etc.), and short words (less than three characters), and converted all text to lower case. We then lemmatized the remaining words using the R package *textstem* (Rinker 2018b) and the *lexicon* (Rinker 2018a) engine. This combines various versions of a word ("fall", "fell" and "falling") into one instance of that word ("fall"). This is preferable to stemming, which removes common endings to words but can miss certain peculiarities of the English language.

The process for creating bigrams and transforming proper central bank names to "central bank" is the same as for the OECD Economic Outlooks.

**Appendix B. Creating the OECD Dictionary**

The OECD dictionary was created from the raw OECD text from 1967 to 1980, allowing us to create a dictionary of words entirely void of any time leakage, which we then followed throughout the training and testing sets. There are seven countries available during the 1967 to 1980 period: Canada, France, Germany, Italy, Japan, United States, and the United

Kingdom. When creating our dictionary, we required that a term be present across these seven countries during this time period to avoid including country-specific words in the text. In addition, we required that the tokens be mentioned at least 50 times over the entire time period across all countries, in order to remove infrequent or misspelled words along with words very specific to a single document. This provided us with a list of terms, from which we manually removed four types of terms: (1) references to time ("year", "summer", "January"), (2) units ("percent", "billions"), (3) section pointers ("outlook", referencing OECD Economic Outlooks, "source"), (4) common typos ("tion", "ing", as words are often split and these common endings may be picked up as unique terms).

We were then left with a final dictionary of 881 terms.

## Notes

[1]  Certain types of crises are determined by thresholds based on market data. For example, thresholds related to currency depreciation can be used to define a currency crisis, as in Laeven and Valencia (2013), and thresholds related to a fall in bank stock prices can be used to define a banking crisis, as in Baron et al. (2020). However, these thresholds are at times somewhat arbitrary, and in general financial crises are usually determined in a narrative fashion.

[2]  For the other 10 countries, realized volatility data are available after 1981 for the following countries (with the year data become available indicated in parentheses): Austria (1985), Belgium (1985), Finland (1987), Greece (1988), Ireland (1986), Iceland (1993), Luxembourg (1985), Norway (1991), Portugal (1986), and Turkey (1987).

[3]  Please refer to Appendix A for information on the precise way in which we gathered and cleaned the textual data.

[4]  Reinhart and Rogoff (2009) argues that Mexico's banking crisis lasted until 2000.

[5]  We only implemented the model on major or more severe crises due to low data availability for the most severe crises.

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
