# Peer review of "Identifying Financial Crises Using Machine Learning on Textual Data"

_jrfm, doi:10.3390/jrfm16030161_

Round 1

Reviewer 1 Report

The paper provides an interesting approach to the detection of financial crisis based on advanced machine learning techniques and a variety of data sources.

I have spotted a couple of typos (eg. Reciever) so the authors should perform a careful review prior to the paper publication.

Recommendation: It would be beneficial for the authors to provide a clearer view on how the detection method they have developed can be exploited as a decision making tool in the hands of policymakers. The point refers to whether the method provides some information on economy crisis that the policy makers do not already have in order to decide for appropriate measures.

Reviewer 2 Report

Summary of the paper:

This study focuses on the financial crisis identifying, establishes different models based on three text information, including OECD text, TRNA text and IMF Article IVs, meanwhile analyzes the effectiveness of these models, especially the text model, and discusses the effect of different texts on crisis recognition. The research finds that text data helps reduce the error rate of models.

Main review:

1. In terms of data, what is the source and time range of volatility and credit data in the article? According to Figure 4, the text document used in this paper has some characteristics, which are helpful for real-time monitoring and prediction of financial crisis. What are these "characteristics"? Why is it helpful to predict? I recommend to add a brief description to this.

2. In the "Machine Learning on Textual Data" section: (1) Paragraph 3 describes the final feature matrix obtained by normalizing the contents in the blue box of Figure 5, and in paragraph 4, the text information is divided from the time dimension for training and testing the machine learning model. How are these feature matrices used in the model? I recommend explaining the use of text information from the perspective of model expressions. (2) The last paragraph states modeling logistic regression using the credit-to-GDP gap as one of regressors. What are the other variables? I suggest to write the expression of the model and a brief description of the variables used in it.

3. In the results part: (1) What does the word "Average Model" mean? The article indicates that according to Figure 8, these models transmit different information when identifying crises. What are the differences? It would be better to provide further clarification. (2) The paper finds that the band of the lower left figure in Figure 18 is narrower than that of the other three figures, but there is little difference between the lower left figure and the upper right figure, so I recommend to further improve it and use (a), (b), (c) and (d) to represent the sub figure. (3) According to Figure 21, the prediction performance of the credit model is considered the best. How can we draw the conclusion? Why does this phenomenon appear? It would be better to provide further clarification.

4. What are the indicators in Figure 1? The specific date referred to by "present" in Figure 1 and Figure 9? Figure 9 and Figure 12 do not provide explanation and analysis. I recommend to provide further clarification. Still, the color difference in Figure 3 is not obvious, it would be better to further improve.

5. Some details: (1) The international organization OECD and IMF need to be presented by their full name when they are first mentioned (Page 2, Line 10 from the bottom). (2) The format of the reference needs to be uniform. Some references use italics to indicate the source of documents, while others use italics to indicate the title of reference, and some do not write the journal, such as the last one.

Reviewer 3 Report

The paper talks about the use of machine learning techniques on textual data to identify financial crises. The paper is well written, really understandable, and useful to understand the procedures to obtain the vocabolary and the relative machine learning approaches. The paper is interesting, not really original (there is a lot of scientific literature devolved to this argument) but the developed techniques are well perform the nowcasting, forecasting, and backcasting. This is confirmed by the high values of auroc. The paper is ready for publication.

Round 2

Reviewer 2 Report

The authors have addessed all my concerns.